# Characterization of Ultrafine Particles and VOCs Emitted from a 3D Printer

**DOI:** 10.3390/ijerph18030929

**Published:** 2021-01-21

**Authors:** Sarka Bernatikova, Ales Dudacek, Radka Prichystalova, Vit Klecka, Lucie Kocurkova

**Affiliations:** 1Department of Fire Protection, Faculty of Safety Engineering, VSB—Technical University of Ostrava, CZ708 00 Ostrava, Czech Republic; ales.dudacek@vsb.cz (A.D.); vit.klecka@vsb.cz (V.K.); 2Department of Occupational and Process Safety, Faculty of Safety Engineering, VSB—Technical University of Ostrava, CZ708 00 Ostrava, Czech Republic; radka.prichystalova@vsb.cz (R.P.); lucie.kocurkova@vsb.cz (L.K.)

**Keywords:** ultrafine particles, 3D printing, VOCs, exposure

## Abstract

Currently, widely available three-dimensional (3D) printers are very popular with the public. Previous research has shown that these printers can emit ultrafine particles (UFPs) and volatile organic compounds (VOCs). Several studies have examined the emissivity of filaments from 3D printing, except glycol modified polyethylene terephthalate (PETG) and styrene free co-polyester (NGEN) filaments. The aim of this study was to evaluate UFP and VOC emissions when printing using a commonly available 3D printer (ORIGINAL PRUSA i3 MK2 printer) using PETG and NGEN. The concentrations of UFPs were determined via measurements of particle number concentration and size distribution. A thermal analysis was carried out to ascertain whether signs of fiber decomposition would occur at printing temperatures. The total amount of VOCs was determined using a photoionization detector, and qualitatively analyzed via gas chromatography-mass spectrometry. The total particle concentrations were 3.88 × 10^10^ particles for PETG and 6.01 × 10^9^ particles for NGEN. VOCs at very low concentrations were detected in both filaments, namely ethylbenzene, toluene, and xylene. In addition, styrene was identified in PETG. On the basis of our results, we recommend conducting additional measurements, to more accurately quantify personal exposure to both UFPs and VOCs, focusing on longer exposure as it can be a source of potential cancer risk.

## 1. Introduction

Currently, three-dimensional (3D) printers are becoming commonplace in offices and libraries as well as in private homes to produce a physical model or product from a digital master pattern. This technology synthesizes three-dimensional objects using an additive method, in which thin layers of material are successively deposited [1]. The printing process uses thermoplastic filaments that are extruded through a high temperature nozzle [2,3]. The extruded material is immediately cooled, and then hardens forming a solid layer [1,4]. This printing technique is derived from Fusion Deposition Model (FDM^TM^) and is one of the most popular, simplest, and cheapest technologies used [1]. Only thermoplastic filaments can be processed with FDM. A variety of polymeric materials are used in 3D printers, such as polylactid acid (PLA), glycol modified polyethylene terephthalate (PETG), acrylonitrilbutadienstyrene (ABS), high impact polystyrene (HIPS), polyvinylalcohol (PVA), and styrene free co-polyester, NGEN, based on an Eastman Amphora™ AM3300 3D polymer. The filament can also be composed of materials which consist of a main plastic base and a second material in the form of dust (woodfill, copperfill, bronzefill, glow-in-the-dark, carbon, or aramid composites, and many others).

It is known that thermoplastic extrusion can emit various hazardous compounds into the air, depending on the material used, for example, polycyclic aromatic hydrocarbons (PAHs), volatile organic compounds (VOCs), and acrylonitrile [1,3]. Previously published studies have proven that some materials used for 3D printing by the FDM method including ABS, PLA, PVA, and HIPS emit ultrafine particles (UFPs) and VOCs (for examples see [1,5,6,7,8,9,10,11,12,13,14,15]). The increasing number of hours of 3D printing in household living areas raises the issue of possible emissions of UFPs and VOCs from printers that use polymeric materials. To pay attention to emissions from 3D printers is important based on past experiences with office equipment as a possible source of air pollution [3,4,16] and the potential negative impacts on human health. Emerging evidence has suggested that inhalation of emissions from material extrusion is associated with adverse effects on the cardiovascular system and respiratory tract [17,18]. A case has already been reported where occupational exposure to ABS filament emissions resulted in asthma [19,20].

The likely exposure to ultrafine particles from 3D printing is a complex issue because 3D printers, commonly used at the consumer level, are not all provided with control measures in terms of exposure and those with emission reduction methods applicable for consumer use are limited and ineffective for reducing exposure levels especially for VOCs [13]. In addition, these printers are widely used due to their affordability and the physicochemical composition of emitting compounds can vary greatly depending on the material used.

The filaments most commonly investigated are PLA, ABS, HIPS, and NYLON. The commonly detected VOCs from 3D printing that have been reported are styrene, caprolactam, ethylbenzene, xylenes, aldehydes, and others [13]. Styrene has been found from ABS and HIPS filaments [6,9,11,12,13], caprolactam from nylon-based filaments [6], ethylbenzene, xylenes and aldehydes from ABS [4,6,9,12], acetone, as well as formaldehyde and toluene from PETG [11,12]. Currently, the widespread use of PETG and NGEN filaments has not been sufficiently investigated. We identified only two studies [11,12] that evaluated the emission of UFPs and VOCs from PETG filament. No studies were found reporting the release of UFPs and VOCs during printing with printers using NGEN filament.

The aim of this study was to evaluate UFPs and VOCs emissions when printing on a commercially available 3D printer with PETG and NGEN. PETG is used in a variety of signage, packaging, industrial, and medical applications, for example, medical braces, bottles, and electronics [21]. The use of NGEN is similar to PETG with a wider range of usage from prototypes and aesthetic models to high-tech prosthetics [22]. These materials can be easily used for printing various objects in the home environment, where control measures, in terms of exposure, are not usually applied. Experimental measurements were performed to evaluate whether, and in what amounts, the use of PETG and NGEN filaments in 3D printers releases UFPs and VOCs into the air.

## 2. Materials and Methods

### 2.1. Description of the Particle Source: Printer and Filaments

UFP and VOC emissions were measured during 3D printing using an ORIGINAL PRUSA i3 MK2 printer. The layout of the printer in a glass box is shown in Figure 1. For measurement purposes, the printer was placed in an enclosed glass box measuring 0.70 × 0.59 × 0.70 m with an internal volume of 0.29 m^3^, without forced air exchange, in order to determine the maximum values of emissions released into the environment during printing. Two small openings made in the walls of the box were used to bring cables into the box and to feed the filaments to the printer and the opening was sealed with aluminium self-adhesive tape. The front of the box was removable for inserting devices and handling the printer. A face glass wall was placed against the opening for handling, and adhesive tape and a flexible strap were used to secure it against the opening. The printer box was placed on a table in the center of the room. The electrostatically conductive tabletop was connected to the laboratory earthing system. Aluminium self-adhesive tapes used on the chamber were also connected to this plate to balance any electric charge. Sealing of the chamber was verified after the chamber was constructed and, subsequently, after any modification of cable passages, etc. by a leak test, i.e., filling the chamber with isobutylene (reaching a concentration around 10 ppm) and detecting possible leakage out of the chamber using a photoionization detector (PID), i.e., RAE Systems ppbRAE 3000, produced by RAE Systems, San Jose, CA, USA (now Honeywell, Safety and Productivity Solution), purchased from Chromservis s.r.o., Prague, Czech Republic.

Two different filaments of similar characteristics, namely PETG (polyethylene terephthalate glycol-modified) with a diameter of 1.75 mm and NGEN (co-polyester, 3D Polymer Eastman Amphora™) with a diameter of 1.75 mm, were used for printing. Both filaments were of orange color and supplied by Filament PM, Haňovice, Czech Republic (PETG), and Prusa Research, Praha, Czech Republic (NGEN). The basic functional characteristics of the measured filaments are given in Table 1. The properties are provided by the PETG and NGEN producers and should be looked at critically. Both of the selected filaments feature properties of a high range of temperature resistance and resistance to chemicals such as solvents and weak solutions of acids and bases. This material can be printed on printers without a heated chamber or printing plate. 

The printer was programmed to print a cube-shaped block with notches, inside with infill (see Figure 2) measuring 43 × 43 × 43 mm. The weight of the printed cube using PETG filament was 30.5 ± 0.5 g, and 34.8 ± 0.5 g using NGEN filament. The time to print one test cube after heating the nozzle and the base plate to the desired temperature using PETG filament took approximately 1 h, and approximately 1 h and 21 min using NGEN filament.

### 2.2. Strategy and Method of Measuring Ultrafine Particles (UFPs)

A particle concentration monitor was placed next to the 3D printer. Before printing, the background value was measured for 15 min, then, the printer was turned on and warm-up was set. The nozzle and the base plate heating temperatures, t, were 245/75 °C and 230/85 °C, when using PETG filament and NGEN filament, respectively, as recommended by the manufacturer. Once the heating phase was completed, printing was started, and the glass box shut.

Particle number measurements were performed at 1 s intervals using a condensation particle counter (CPC model 3007, TSI Inc., USA). The CPC 3007 measures the total particle number concentration with a size of 10–1000 nm and has a concentration range of 10^5^ particles per cm^3^. During printing, particle size distribution was also measured using a scanning mobility particle sizer spectrometer (SMPS model 3910, TSI Inc., USA). The SMPS can sort and count particles measuring in the size range 10–420 nm into 13 size channels. All measuring instruments used had a valid calibration performed by TSI Instruments LTD and were checked by daily inspection before each measurement. Antistatic tubing, free of sharp bends, was used for sampling with the length less than 0.5 m to minimize particle losses [24]. The inputs of the sampling probes were placed about 10 cm from the printer.

In addition to the actual measurement of particulate emissions during cube printing, the so-called “zero test” was performed during the printing simulation. This test was performed because a release of particles was expected at the warm-up stage of the metal surfaces [25], i.e., during the thermal process of heating the nozzle and the base plate to the recommended printing temperature. These particles release likely causes higher emission values than during the actual printing [1]. The “zero test” was conducted during printing in the absence of a filament but at the same printer setting as in the case of actual printing for following comparison.

Particle number concentration and size distribution were measured during all phases of printing (heating, printing, cooling), including 15 min of background sampling before heating and printing was started. The total time measured during one printing task ranged between 1:15 and 1:40 h, depending on the chosen filament. The measurements were performed for each printing task using PETG and NGEN (the cube was printed 3 times per each filament). A summary of the parameters of the tests performed are shown in Table 2. 

The measurements were supplemented by values of the surrounding microclimatic conditions measured in the laboratory and in the glass box. While the laboratory conditions were relatively stable, with a temperature of 21.5 ± 0.8 °C and relative humidity of 33.4 ± 2.2%, the glass box temperature increased and its relative humidity dropped, during the printing task with PETG by +10 °C and −11%, respectively, and +14 °C and −13%, respectively, during the printing task with NGEN.

The obtained data were processed by basic statistical analysis in software programs of individual measuring instruments, namely NanoScan Manager Software and Aerosol Instrument Manager Software. Subsequent processing including graphs were performed by the Python programming language, version 3.7.

Particle emission rates (*PER*) and total particle emissions (TP) were calculated based on the UL2904, the Standard Method for Testing and Assessing Particle and Chemical Emissions from 3D Printers [26]. Particle concentrations (*C_p_*) were calculated and particle number concentrations reported by the CPC (D_p_ > 7 nm) were averaged over nominally 1 min to smooth the data.

The particle loss coefficient (*β*), needed for calculation of *PER*, was calculated based on the exponential decay of particles after printing stopped, as follows:(1)β= lnc1c2t2−t1,
where *t*_1_ (s) is at least 5 min after the end of the print phase and *t*_2_ (s) is at least 25 min after *t*_1_. *C*_1_ (#/cm^3^) and *C*_2_ (#/cm^3^) are the corresponding particle concentrations. The unit for *β* is s^−1^ for number concentration.

Particle emission rates (*PER*) as a function of time were calculated using *C_p_* (corrected with *β*) as:(2)$$PER=V\;\left(\frac{c_{p}\;\left(t\right)-c_{p}\left(t-\Delta t\right)\exp\left(-\beta \cdot \Delta t\right)}{\Delta t\;\exp\left(-\beta \cdot \Delta t\right)}\right),$$
where *V* (cm^3^) is the volume of the chamber and Δ*t* (s) is the time interval between two successive data points. The unit for *PER* is #/s for number concentration.

Total particle emissions (*TP*) for the complete print job were calculated by integrating particle concentrations over the emission period, which was determined from the *C_p_* and *PER* curve as:(3)$$TP=V\;\left(\frac{\Delta c_{p}}{t_{stop}-t_{start}}+\beta \cdot c_{av}\right)\left(t_{stop}-t_{start}\right),$$
where *t_start_* is the time when *C_p_* begins to increase, *t_stop_* is when *PER* remains steady (below 10% of the maximum of *PER* over at least the next 10 min), Δ*C_p_* (#/cm^3^) is the difference in *C_p_* between *t_stop_* and *t_start_*, and *C_av_* (#/cm^3^) is the arithmetic average of *C_p_* between *t_start_* and *t_stop_*. The unit for *TP* is # for number concentration.

### 2.3. Thermal Analysis

Thermal analysis of PETG and NGEN filaments was performed to determine the temperatures at which decomposition and changes in filament properties occurred. Filament decomposition is accompanied by changes in the weight of the filament sample or an exothermic or endothermic process therein. If these changes occur in the temperature range suitable for 3D printing (i.e., extruder temperatures between 220 °C and 270 °C), significant emissions from 3D printing can be expected, among other things. The tested properties were weight loss, as well as and exothermic and endothermic processes in filaments. Thermogravimetric analysis (TGA) and differential scanning calorimetry (DSC) methods were used simultaneously for the thermal analysis of the filaments using a Mettler-Toledo TGA/DSC 2 instrument in the temperature range 25–750 °C, with a temperature rise rate of 20.00 K/min and an air flow rate of 50.0 mL/min, also using an Alumina crucible with a volume of 70 µL.

### 2.4. Quantitative and Qualitative Determination of Volatile Organic Compounds (VOCs)

Three-dimensional (3D) printing of the test body (cube) was performed for the quantitative determination and qualitative analysis of VOCs released during printing. As compared with the printing for UFP determination, the printing program was modified so that the printing time was the same for both filaments, approximately 1 h. The changes concerned the printing properties (optimization) and the printing geometry of the cube (infill type, wall thickness, etc.) were the same. For each combination of filament material and extruder temperature, one print job was performed for quantitative and qualitative determination of VOCs. Printings were made in the extruder (nozzle) temperature range between 220 °C and 270 °C, with a stepped increase of 10 °C; the base plate temperature at all printing was 85 °C. To cover possible non-standard extruder temperature settings by the user, measurements were conducted at a temperature range greater than the filaments manufacturers’ recommendations (Table 1).

In the quantitative analysis, the total amount of VOCs and the time changes of the VOC concentrations in the test chamber were determined during printing. The photoionization detector (PID), RAE Systems ppbRAE 3000, is directly equipped with a datalogger, which was used for the measurements. A standard 10.6 eV lamp was used for the measurements, and the correction factor expressing the detector response to individual VOCs was set to CF = 1. The PID detector was placed directly inside and at the bottom of the test chamber. A suction tube, for air samples, was placed 120 mm above the bottom of the chamber, and the measured VOC concentrations were recorded by the built-in data logger, at an interval of 1 s.

A GC-MS (gas chromatography-mass spectrometry) FLIR GRIFFIN 460 was used for qualitative analysis. A preconcentration tube with thermal desorption was used to take air samples from the test chamber with the 3D printer. Two preconcentration tubes with thermal desorption (each with TENAX-TA and CARBOXEN 1017) are a direct part of the GC-MS GRIFFIN 460. During the qualitative analysis, the universal sampling port GC-MS was connected to the test chamber with a Teflon hose with an outer diameter of 6 mm. The exhaust point was in the middle of the test chamber. The flow rate of the samples taken through the Precon tubes was set at 350 mL/min, and the sampling time was 5 min. Areas on the total ion chromatogram (TIC), below the Quant Ion, were used as a guide to the relative quantitative expression of the concentration of compounds identified by GC-MS. For each identified compound, the relative increase in concentration was determined by subtracting the initial area corresponding to that compound before starting 3D printing, from the area corresponding to that compound on the TIC after 3D printing was completed.

## 3. Results

### 3.1. Particle Number Concentrations of UFPs

A common result of both filaments after printing begins is a sharp increase in the emissions of the 3D printer. It is usually the maximum measured value of the concentration recorded during the measurement of the entire printing task, which corresponds with the results of already published works [4,6,7,10,27]. This is in line with the formation of newly generated particles in the vicinity of the nozzle, as a consequence of the molten fiber. It can contain organic compounds and other related types from the filament bulk form or additives. Figure 3 depicts the resulting time-resolved measured values for the “zero test”, PETG and NGEN filament printing, during all phases of the entire printing task. It is possible to monitor the development of the total particle number concentration measured by CPC 3007 over time, i.e., starting from the measurement of the background values to measurements during the heating and printing, up to the cooling of the printer. The graph shows a small increase in concentration during the heating phase, and then a sharp increase in concentration at the start of printing. It could also be influenced by closing the box, thus, prohibiting exchange of air inside the box with the surrounding air.

To allow a better comparison of emissions released during the individual operations, we divided the values by the individual phases of the experiment, namely the background, heating, printing, and cooling phases (see Figure 4). The average and maximum values of size differentiated concentrations are presented in Appendix A (see Appendix A).

From Figure 3 and Figure 4, it can be seen that the background concentrations during measurements were analogous and their values ranged up to 5000 particles per cubic centimetre, which are quite normal values in a relatively clean environment. As compared with the background, an increase in concentrations was observed during heating of both filaments. In these cases, it was not possible to identify the source of the unusual short-term increase in the number of particles (given by the standard deviation of the measured data). Attempts to identify what operation the printer performed during these two measurements (as opposed to the other measurements) failed. 

Figure 3 and Figure 4 also show the rise in the concentrations at the start of the printing tasks (as compared with the background), the increase with PETG filament is, on average, about 10 times greater than the increase in the concentrations when printing with NGEN filament. Increased concentrations were also measured during the cooling down of the printer and printed sample. 

The average PERs during printing, for PETG and NGEN, are shown in Figure 5. The TP calculated for the total particle concentrations were 3.88 × 10^10^ particles for PETG and 6.01 × 10^9^ particles for NGEN. The relationship between overall emission and print object were compared with other published works on 3D printers. Therefore, the determination of yields were estimated yields from each print job, using the ratio of TP over object mass [9]. These resulting values are 1.28 × 10^9^ particles for PETG and 1.73 × 10^8^ particles for NGEN, per 1 g on the printed objects.

By monitoring the size distribution, it is possible to observe an increase in concentrations of particles with a median of 20 nm and 11.5 nm mode in the first minutes of the start of printing with PETG, or a median of 50 nm and 48.7 nm mode at the start of the printing with NGEN. Then, the median shifts, over time, towards larger particles due to their coagulation. The size distribution of particles is presented in Appendix A. These results confirm the development of particles smaller than 100 nm during the printing using both filaments. The most abundant size of the particle number concentration in the sampling reaches values of 6 × 10^5^ #/cm^3^ during the printing with PETG, or 7 × 10^4^ #/cm^3^ during the printing with NGEN. 

### 3.2. Thermal Analysis

The result of the performed thermal analysis of the PETG and NGEN filament samples is shown in Figure 6.

Observable weight loss accompanied by endothermic decomposition (or endothermic decomposition with heat release by oxidation processes) begins above 330 °C. A decrease of approximately 90% in weight occurred in the temperature range between 330 and 504 °C with PETG filament and between 330 and 450 °C with NGEN filament. The results of the thermal analysis show that no significant emissions of VOCs into the air can be expected in the temperature range between 220 and 270 °C.

### 3.3. Determination of the Total Amount of VOC

The VOC concentrations in the test chamber at different printing temperatures are presented in Figure 7 for NGEN and Figure 8 for PETG filaments. The time 0:00:00 in the figures corresponds to the time when a measurable VOCs concentration appeared in the test chamber.

It is evident from Figure 6 and Figure 7 that the total VOC concentration is dependent on the temperature of the extruder. The total achieved VOC concentrations were relatively low; the total VOC concentration with PETG filament was 550 ppb and with NGEN filament did not exceed 600 ppb. The low concentrations correspond to the conclusion of the thermal analysis of both filaments.

Because the 3D printer is a significant source of heat, the temperature rose in the test chamber during the measurement about 5 to 7 °C from the original value. The temperature rise in the test chamber was lower than during the measurements of UFP, due to the use of a lower printing base plate temperature. The temperature in the test chamber increased as the temperature of the extruder increased. The temperature of the actual printer also rose in the course of printing, in particular, the temperature of its electric motors, control electronics, and the like. Considering the fact that it was a new printer, it is possible to assume that the total VOC concentration outside the VOCs of the filament also involved VOCs emitted from some part of the printer that was warming up.

### 3.4. Qualitative GC-MS Analysis

The results of a qualitative analysis of 3D printing are presented for extruder temperatures between 220 and 270 °C. Figure 9 shows an example of a TIC analysis of NGEN filament after the completion of the 3D printing with the extruder temperature at 240 °C. 

The dominant compounds during printing with PETG filament at all printing temperatures are xylene, toluene, and ethylbenzene; ther compounds are 1-octanol, trimethylbenzene, nonanal, napthalene, and decanal. Benzene appears at the extruder temperature of 250 °C. The dominant compounds during printing with NGEN filament present at all printing temperatures are xylene, toluene, and ethylbenzene. Nonanal, napthalene, decanal, benzene 1-ethyl-2.4-dimethyl- and trimethylbenzene are present at all temperatures as well. The presence of benzene appeared, in the case of PETG, at the extruder temperature of 250 °C. The results of GC-MS analysis are presented in Appendix A.

A comparison of the qualitative analysis with NGEN and PETG filaments is shown in Table 3 (YES indicates the identification/presence of the given compound in the analysis of the air in the test chamber at the indicated extruder temperature). The comparison of relative values of Quant Ion intensity rises with the increasing extruder temperature are presented in Appendix A (see Appendix A).

While many compounds occur in both filaments and at different printing temperatures (toluene, ethylbenzene, zylene, nonanal, napthalene, and decanal), 1-octanol was identified only in PETG filament at all extruder temperatures. In four of the six cases, styrene was also identified in PETG filament. Benzene occurs mainly at higher printing temperatures, with the exception of printing with PETG filament at 220 °C, where benzene was identified with a relative Quant Ion intensity of 1.6 times lower than at a temperature of 250 °C. Trimethylbenzene was identified in all analyses (1,2,4-trimethylbenzene in eleven analyses and 1,3,5-trimethylbenzene in nine analyses). According to the results of the analyses, the occurrence of xylene, toluene, and ethylbenzene can be considered to be dominant. With respect to VOC determination, it is not possible to distinguish in a qualitative GC-MS analysis if the compound was released from filaments or released from the individual components of the 3D printer during printing.

## 4. Discussion

During printing, VOC emissions using both filaments were very low; the total VOC concentrations in the test chamber were in the order of hundreds of ppb after one hour of printing. When evaluating, it is necessary to take into account both the small volume of the test chamber (0.29 m^3^) and the printing time of about 1 h, which is rather at the lower limit of printing times for 3D printing using printers of this type. In addition to the heated filament, the heated components of the 3D printer can also be a source of VOCs in 3D printing. The release of VOCs from the filament can be expected both at its outlet from the extruder (where its temperature is highest), from the surface of the currently printed sample layer with a temperature close to the extruder temperature, and then from the remaining surface of the gradually cooling printed sample. Because these parameters are not constant or linearly varying, a constant or linearly varying rate of VOC release cannot be expected during printing. A conservative approach was used to estimate the VOC concentration for an 8-h shift in a room with an area of 115 m^3^ without considering ventilation and infiltration, where a single printer would be operated. A simple scaling method was used. The method was based on the average VOC release rate in the chamber, when printing at an extruder temperature of 270 °C from the fifth to the tenth minute, when the transient effects of the printing initiation no longer applied and there was no significant decrease in the VOC release rate over time (see Figure 6 and Figure 7). The estimated VOC concentration in the room in this case could be up to 16.4 ppb for PETG and up to 19.4 ppb for NGEN. 

A chemical analysis of the composition of released particles during printing was not performed, because of the low concentration of particles and the limitation of the volume of sampling air during the performed measurements. For particles at very low concentrations, a high volume of the sample is preferred, but this is difficult to achieve in a chamber-type experiment, because the flow rate for taking a sample is limited by the size of the chamber and air-change rate and the relatively short printing time limits the time of the sampling.

Given that previous studies have focused mainly on PLA and ABS, a limited amount of data is available for comparison. UFP sampling of PETG printing in the Gu et al. study (2019) was performed using FMPS (particle size range 5.6–560 nm) and OPS (particle size range 0.3–10 μm), which was a much wider range than determined in this study. The UFP sampling was performed with a CPC 3007 instrument for particle sizes in the range of 10–1000 nm. Furthermore, [11] used a different color of filament, namely PETG black, where the dye used could also affect the identified compounds and the resulting VOC concentrations.

The UFP measurement of emissions and concentrations of VOCs that are listed, here, are also important for assessing the possible impact of exposures on a person’s health. For example, ethylbenzene, which the International Agency for Research on Cancer (IARC Group 2B classification) classified as a possible human carcinogen [28,29], was emitted when printing with PETG and NGEN fibers. Styrene (IARC 2A classification group) was also emitted using a PETG fiber [29]. The concentration of total VOC in this study was 550 ppb for PETG and 600 for NGEN. The short-term exposure limit for total VOCs, published by the World Health Organization, is 100 ppb for long-term exposure [30]. If a measured concentration was converted to long-term exposure, the concentration would be under this threshold. However, caution should be taken when the printing continues for several hours.

To provide a basis for comparison with regulatory exposure limits and to help to understand the potential consequences for human health, we would need to make estimates of UFP and individual VOC emissions to predict steady state concentrations likely to result from constant printer operation in a conventional office environment. This work does not serve as a detailed exposure model, but rather as a screening analysis of the quantity of UFP and VOCs that users of 3D printers can be exposed to. Although there is considerable uncertainty in the estimation of the amount an employee may be exposed to, exposure to UFP and VOCs from desktop 3D printing in a typical office or home environment during long regular printing using a PETG or NGEN filament could lead to adverse health consequences, especially, for sensitive individuals.

## 5. Conclusions

On the basis of the obtained data, we have proven that exposure can occur during actual printing and cooling. During the tests conducted in the chamber, there was a greater emission of particles during printing tasks with PETG filament. It should not cause any harm to human health within consumer use, if the exposure is short term and occasional. Higher levels of exposure could be expected within nonproductive applications in educational institutions with occasional longer printing times, for example, printing of teaching tools. Although we are not aware of any regulatory limits for indoor UFP concentrations, increasing UFP concentrations to ~146,000 cm^−3^ could be a value many times higher than that usually observed in indoor air in office and laboratory environments. Although the measurements showed that VOCs were present in the indoor air, the concentration was low for short-term exposure. Nevertheless, caution should be taken in the case of exposure to carcinogens such as no threshold substances. The following recommendations are based on these findings: Additional measurements should be conducted for more precise quantifications of personal exposure to UFP, as well as VOCs. Furthermore, the producers should try to design a printing technology with a low emission rate. If such a technology cannot be developed, efforts should be made for technical measures that would restrict UFP and VOC emissions, for example, by enclosing machines, providing local exhaust ventilation, and introducing combined gas and particle filtration systems. For personal printing in households, it is recommended to print in a space without the presence of persons and to ventilate the space.

## Figures and Tables

**Figure 1 ijerph-18-00929-f001:**
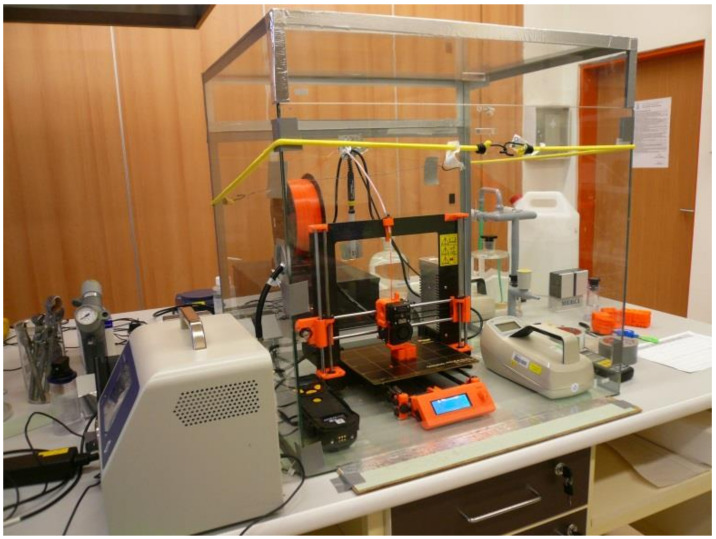
Glass box with printer and measuring instruments.

**Figure 2 ijerph-18-00929-f002:**
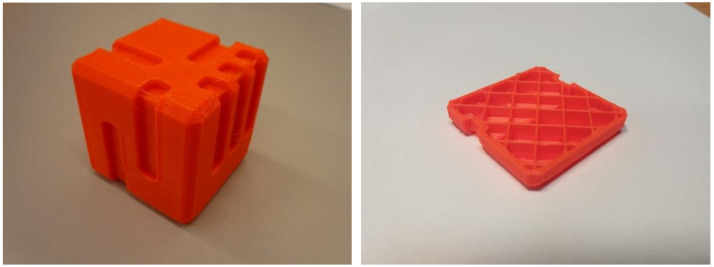
Printed cube and a cross-section of the printed cube.

**Figure 3 ijerph-18-00929-f003:**
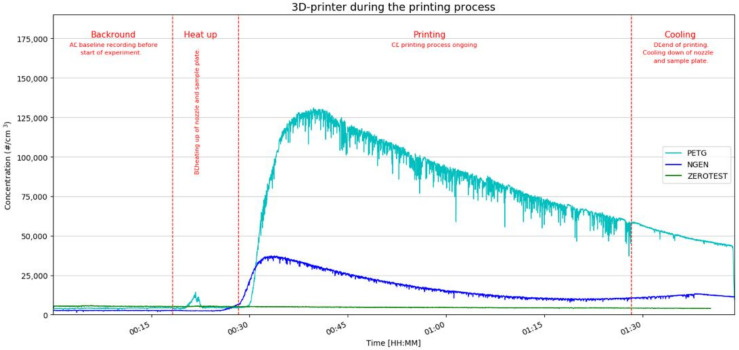
Values of the total particle number concentration during printing with glycol modified polyethylene terephthalate (PETG) + styrene free co-polyester (NGEN) filaments + zero test.

**Figure 4 ijerph-18-00929-f004:**
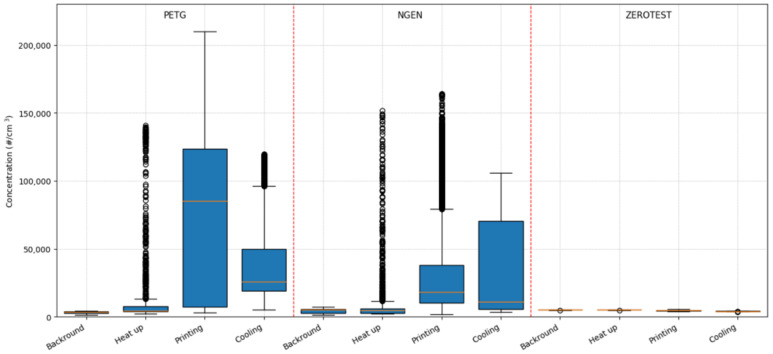
Particle number concentration emitted by polyethylene terephthalate (PETG) and styrene free co-polyester (NGEN) filaments and the zero test.

**Figure 5 ijerph-18-00929-f005:**
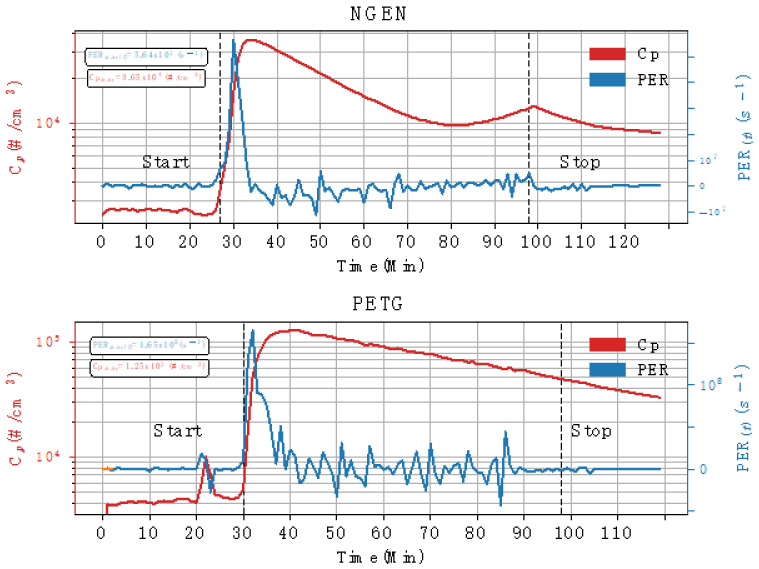
Average particle emission rates (PERs) during printing with polyethylene terephthalate (PETG) and styrene free co-polyester (NGEN).

**Figure 6 ijerph-18-00929-f006:**
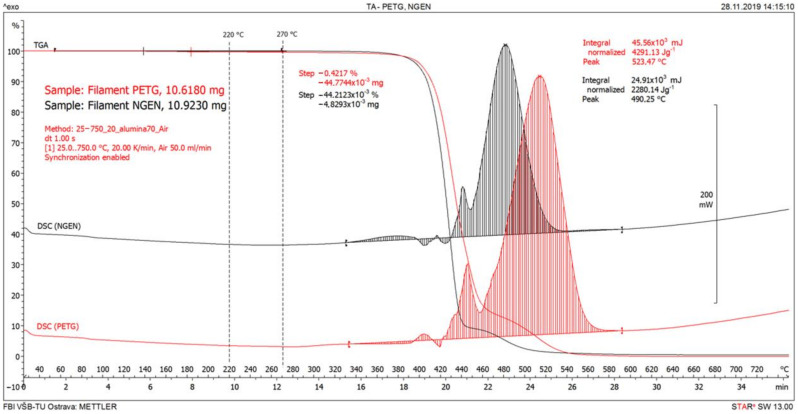
Thermal analyses with polyethylene terephthalate (PETG) and styrene free co-polyester (NGEN) filaments.

**Figure 7 ijerph-18-00929-f007:**
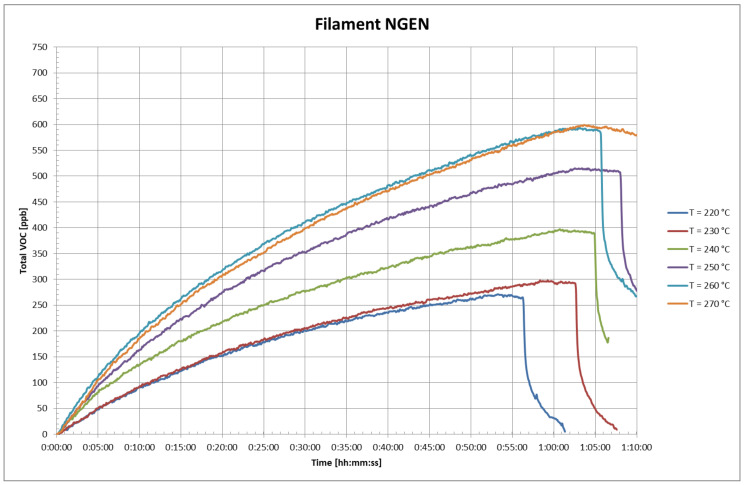
Total volatile organic compound (VOC) concentrations in the test chamber in the case of styrene free co-polyester (NGEN) filament.

**Figure 8 ijerph-18-00929-f008:**
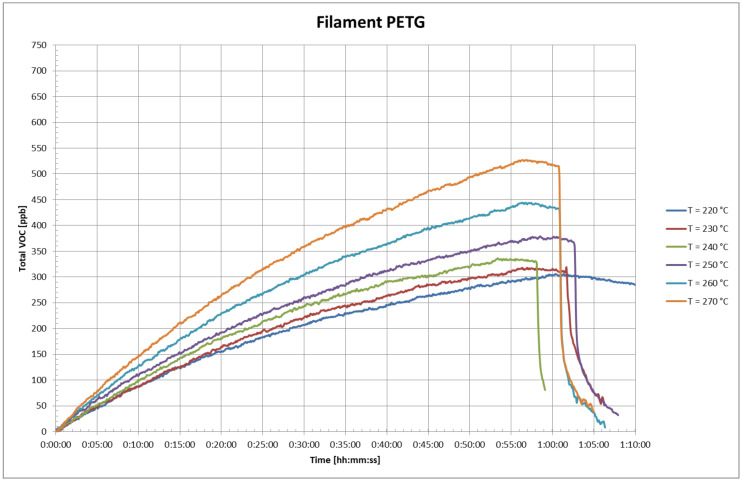
Total volatile organic compound (VOC) concentrations in the test chamber in the case with polyethylene terephthalate (PETG) filament.

**Figure 9 ijerph-18-00929-f009:**
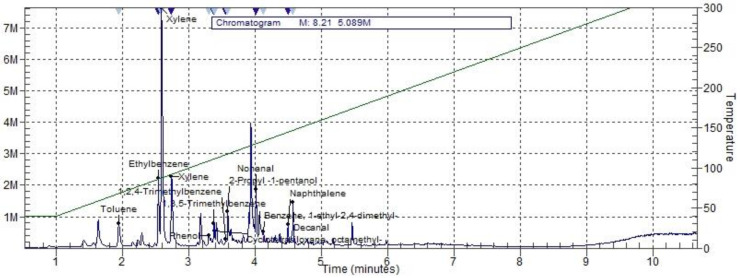
Example of a total ion chromatogram (TIC) with styrene free co-polyester (NGEN) filament.

**Table 1 ijerph-18-00929-t001:** Comparison of properties of filaments used [23].

Filament Properties	NGEN	PETG	PLA	ABS
Glass temperature (°C)	85	75	50	100
Toughness	++	+++	–	+++
Printing temperature (°C)	220–240	240–260	190–220	250–260
Printability	+++	++	+++	+
Odour neutral (during printing)	+++	++	+	–
Stability during printing	+++	++	+	+
Surface clarity	+++	++	+	–

– worse, ++ good, +++ better.

**Table 2 ijerph-18-00929-t002:** Summary of conditions of ultrafine particles (UFP) tests performed.

Material	Printing Temperature of the Nozzle/Sample Plate (°C)	Filament Length/Weight (m)/(g)	Printing Time (min)	Printing Speed/Layer Height (mm·s^−1^)/(mm)
“Zero test”	245/95	No filament	70	16/0.16
PETG	255/90	9.839/30.266	60	16/0.16
NGEN	230/95	11.62/34.799	82	16/0.16

PETG: polyethylene terephthalate; NGEN: styrene free co-polyester.

**Table 3 ijerph-18-00929-t003:** Comparison of qualitative analysis during three-dimensional (3D) printing with styrene free co-polyester (NGEN) and polyethylene terephthalate (PETG) filaments.

Compound	Temperature											
	220 °C		230 °C		240 °C		250 °C		260 °C		270 °C	
Benzene		YES					YES	YES	YES	YES	YES	YES
Heptane			YES						YES			
Toluene	YES	YES	YES	YES	YES	YES	YES	YES	YES	YES	YES	YES
1-Octanol		YES		YES		YES		YES		YES		YES
Ethylbenzene	YES	YES	YES	YES	YES	YES	YES	YES	YES	YES	YES	YES
Xylene	YES	YES	YES	YES	YES	YES	YES	YES	YES	YES	YES	YES
Styrene		YES				YES				YES		YES
Propylbenzene							YES	YES	YES			
1,3,5-Trimethylbenzene	YES		YES	YES	YES		YES	YES	YES	YES		YES
1,2,4-Trimethylbenzene	YES	YES	YES	YES	YES	YES	YES	YES	YES		YES	YES
Phenol					YES					YES	YES	
Cyclotrisiloxane, hexamethyl-						YES				YES		
Cyclotetrasiloxane, octamethyl-		YES			YES	YES	YES			YES		YES
2-Propyl-1-pentanol	YES	YES		YES	YES	YES		YES	YES	YES	YES	
Nonanal	YES	YES	YES	YES	YES	YES	YES	YES	YES	YES	YES	YES
Benzene, 1-ethyl-2,4-dimethyl-	YES	YES	YES	YES	YES	YES	YES	YES	YES		YES	YES
Decamethylcyclopentasiloxane											YES	
Naphthalene	YES	YES	YES	YES	YES	YES	YES	YES	YES	YES	YES	YES
Decanal	YES	YES	YES	YES	YES	YES	YES	YES	YES	YES	YES	YES

## Data Availability

The data presented in this study are available on request from the corresponding author. The data are not publicly available due to funding of the experiments was not from public sources.

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
