# Peer review of "Characterization of Ultrafine Particles and VOCs Emitted from a 3D Printer"

_ijerph, 2021, doi:10.3390/ijerph18030929_

Round 1
Reviewer 1 Report
Please see attached file.

Author Response
We thank the Reviewer for the positive and thorough reviews. We have made revisions throughout the manuscript which we think have greatly improved the paper. Please see specific responses in Table - part Reviewer 1 (please see the attachment).

Reviewer 2 Report
The manuscript “Characterization of Ultrafine Particles and VOCs Emitted from 3D Printer” describes a study where the emission of ultrafine particles (UFP’s) and volatile organic compounds (VOCs) from two sorts of thermoplastic filaments is evaluated. Despite a similar work has been already published by Azimi et al., the authors have evaluated the emission from PETG and NGEN, which are not used in any previous work. Therefore, this paper could contribute with relevant data concerning 3D printer emissions and it should be published after some minor revisions.
Introduction: There are too many discussions about the negative effect of the emission of UPPs and VOCs by FDM 3D printers, however, the authors did not discuss the chemical composition of such emissions. It should be discussed in-depth (e.g. which chemicals are emitted by each thermoplastic filament).
Table 1: The reference link provided is not working (at least in my browser). Please verify.
Table 2: Please verify the significative figures of the filament weight and length data. It seems to be excessive.
Why the emissions from PETG filament is about 10 times higher than NGEN filament?
A recent paper was published regarding 3D printing compounds evaluation and should be cited (doi.org/10.1016/j.aca.2020.11.012)
Author Response
We thank the Reviewer for the positive and thorough reviews. We have made revisions throughout the manuscript which we think have greatly improved the paper. Please see specific responses in Table - part Reviewer 2 (please see the attachment).

Reviewer 3 Report
By filling the material gap, this paper represents a good contribution to the field, but I think the article needs to do a better job outlining the 3D printing process and describing the material properties. NGEN, for example, is a young material and like PETG a co-polyester. Some examples what these materials can be used for would help the reader to position the material within the fast growing selection of FDM filaments. Table 1, which refers to information given by the producer of NGEN [17], doesn’t provide enough information. In addition to the functional properties, including toughness, also ductility and brittleness are of special interest for the right choice of material application. A rough, unbiased positioning of the two tested materials within commonly used materials, like PLA and ABS, would also help to increase the quality of the article.
Your description of the 3D printing process seems to be heavy-handed. Although “curing” is not generally wrong, it is misleading since there are other 3D printing processes where the material is actively cured with an agent involved. Here it is simple solidification. You should also mention, that only thermoplastic filaments can be processed with FDM.
There are plenty original sources that can be cited for the description of the 3D-printing process. I recommend using those instead of [1,2,3].
The research methods as well as the equipment and test-setup are clearly described. However, it is not explicitly mentioned if the glass door actually has been sealed or allows some leakage.
It should be stated more clearly where the tested PETG comes from. For NGEN there is only one source, but PETG can be bought from different manufacturers who most likely don’t process and treat the material identically, which might have an impact on the results.
As I also mentioned in the manuscript, the inner structure of the printed cube is also of interest and will, due to different surface exposure – influence the results. Not knowing if and what kind of infill has been used, recapitulated tests might deliver different results also due to different inner cube designs.
The results of the UFP-, VOC-measurements and the thermal analysis are plausible and well explained. The supplementary data is valuable and I suggest to place table S2 right within the results section if this doesn’t exceed the limit for the article.
In the discussion section it would be beneficial to compare with emission limits or at least hint to the magnitude which might be harmful for people’s health. This information could help highlighting the impact of your results.
In the conclusion section you assume the exposure of a worker to UFPs and VOCs. Since the printer used is a low-end consumer product you should probably focus on private home use or non-productive application in educational institutions with occasional long exposure times rather than a workplace with continual exposure.
I also recommend going through the article for sentence flow. Overall, it is well written, but there are a few places where the wording and sentence structure can be improved.
Regarding the references, check the Internet-adresses again. E.g. [17] cannot be found: although the information is generally available in the internet, the original source cannot be determined.
There are three more articles, which might be worth mentioning within your scope:
- Zhou , X. Kong , A. Chen , S. Cao , Investigation of ultrafine particle emissions of desktop 3D printers in the clean room, Procedia Eng. 121 (2015) 506–512 .
- Wujtyla , P. Klama , K. Spiewak , T. Baran , 3D printer as a potential source of indoor air pollution, Int. J. Environ. Sci. Technol. 17 (2019) 1–12 .
- Byrley , B.J. George , W.K. Boyes , K. Rogers ,Particle emissions from fused deposition modeling 3D printers: evaluation and meta-analysis, Sci. Total En- viron. 655 (2019) 395–407 .
For the abstract I recommend to mention the type of printer used (consumer FDM), to quantify the VOC emission like you did with the UFP emission and to give a conclusion or statement how these numbers probably impact public health.

Author Response
We thank the Reviewer for the positive and thorough reviews. We have made revisions throughout the manuscript which we think have greatly improved the paper. Please see specific responses in Table - part Reviewer 3 (please see the attachment).

Round 2
Reviewer 1 Report
Thank you for thoughtfully considering my comments.